# Mixtures of Subspaces for Bandwidth Efficient Context Parallel Training

**Sameera Ramasinghe**     **Ajanthan Thalaiyasingam**     **Hadi Mohaghegh Dolatabadi**

**Gil Avraham**     **Violetta Shevchenko**     **Yan Zuo**     **Chamin Hewa Koneputugodage**

**Alexander Long**

Pluralis Research

## Abstract

Pretraining language models with extended context windows enhances their ability to leverage rich information during generation. Existing methods split input sequences into chunks, broadcast them across multiple devices, and compute attention block by block which incurs significant communication overhead. While feasible in high-speed clusters, these methods are impractical for decentralized training over low-bandwidth connections. We propose a compression method for communication-efficient context parallelism in decentralized settings, achieving a remarkable compression rate of over $95\%$ with negligible overhead and no loss in convergence. Our key insight is to exploit the intrinsic low-rank structure of activation outputs by dynamically constraining them to learned mixtures of subspaces via efficient reparameterizations. We demonstrate scaling billion-parameter decentralized models to context lengths exceeding 100K tokens on networks as slow as 300Mbps, matching the wall-clock convergence speed of centralized models on 100Gbps interconnects.

## 1 Introduction

Rapid scaling of large language models (LLMs) has made distributed training a necessity [23, 20, 11, 38]. As both model size and context length continue to grow, efficient training increasingly depends on parallelization across multiple devices. Traditional distributed training paradigms assume high-bandwidth, low-latency interconnects, typically available in centralized data centers. In contrast to such centralized settings, the emerging paradigm of *decentralized training* [53, 39, 24, 22, 21] enables collaborative and democratized machine learning by distributing computation across heterogeneous, geographically dispersed nodes over the Internet, without requiring specialized networking hardware or centralized orchestration.

However, decentralized training presents a core technical challenge: *limited communication bandwidth*. When nodes are connected via commodity networks, communication quickly becomes a bottleneck. Most prior work, has addressed this issue in the context of distributed data parallelism (DDP), where each node maintains a full model replica and synchronizes gradients during training. A variety of bandwidth-efficient techniques, such as gradient quantization [51, 31, 46], sparsification [49, 44, 45], and delayed synchronization [40, 10, 27, 9], have been proposed to reduce overhead in this setting. Pipeline parallelism (PP) [19], where model layers are partitioned across devices, has also been explored to a limited extent [36, 50].

39th Conference on Neural Information Processing Systems (NeurIPS 2025).

A significantly more challenging – and, to our knowledge, entirely unexplored – setting is *context parallelism* (CP) in decentralized environments. CP has become critical for pretraining frontier LLMs, as it enables efficient training with extremely long sequences, often exceeding 100K tokens (e.g., LLaMA 3 [11]: 130K, LLaMA 4: 256K, DeepSeek [26]: 128K, Qwen 3 [52]: 128K), thereby enhancing the model's ability to capture long-range dependencies. In context-parallel training, each node processes a local chunk of the input and broadcasts its attention activations to all other nodes in every layer and at every step. This imposes substantial communication demands, as attention mechanisms require *global access* to all key and value activations. While centralized systems handle this using high-bandwidth interconnects, all-to-all communication becomes prohibitively expensive in decentralized environments with low-bandwidth links.

In this work, we propose a method to *drastically reduce* the communication required for context-parallel attention without sacrificing model quality, enabling decentralized systems connected via standard internet-grade links to match the convergence performance of centralized systems with datacenter-grade bandwidth. Our approach leverages the observation that attention activations (queries, keys, and values) often reside on low-dimensional manifolds. We exploit this structure by factorizing the attention weights so that outputs lie within dynamic mixtures of low-dimensional sub-spaces. To ensure convergence, we optimize the factored weights on a Riemannian product manifold and introduce an efficient reparameterization scheme that significantly reduces computational and communication overhead. Additionally, we provide theoretical guarantees on the expressivity and convergence of our method, offering principled justification for each design choice.

Crucially, this approach introduces only minor architectural changes with negligible training overhead, and these components can be *removed after training*, yielding a standard transformer architecture compatible with existing inference infrastructure and downstream deployment frameworks. We employ our method up to billion-parameter scale models under various settings and demonstrate that our method achieves over **95%** **communication compression** without harming performance, enabling training with long context windows across devices connected via commodity internet (300Mbps), while matching the wall-clock convergence of centralized systems with high-speed interconnects (100Gbps).

## 2 Background and motivation

### 2.1 Context-parallel training and the communication bottleneck

We begin with a brief exposition on CP training and refer the reader to [11] for an extended read. Transformer attention requires each query to interact with all key-value pairs, resulting in a computational complexity that grows quadratically with context window. This becomes particularly prohibitive for long sequences, which makes parallelization strategies essential. In *context-parallel* settings, the input sequence $X \in \mathbb{R}^{n \times d}$, where $n$ is the context length and $d$ is the model dimension, is partitioned across $m$ devices along the context dimension:

$$X = \begin{bmatrix} X_1^\top & \cdots & X_m^\top \end{bmatrix}^\top, \qquad X_i \in \mathbb{R}^{n_i \times d}, \quad \sum_{i=1}^m n_i = n.$$

Each device $i$ computes local queries, keys, and values per head. For clarity we suppress the head index; all quantities are understood to be per attention head unless otherwise stated:

$$Q_i = X_i W_q \in \mathbb{R}^{n_i \times d}, \quad K_i = X_i W_k \in \mathbb{R}^{n_i \times d}, \quad V_i = X_i W_v \in \mathbb{R}^{n_i \times d}.$$

Computing attention locally requires global access to keys and values:

$$K_g = \begin{bmatrix} K_1^\top & \cdots & K_m^\top \end{bmatrix}^\top \in \mathbb{R}^{n \times d}, \qquad V_g = \begin{bmatrix} V_1^\top & \cdots & V_m^\top \end{bmatrix}^\top \in \mathbb{R}^{n \times d},$$

Typically, CP performs (some form of) *all-gather*, where each device broadcasts its local $K_i, V_i$ to form $K_g, V_g$, incurring communication cost $O(nd)$ per device where $d \ll n$. Recently proposed *Ring Attention* [29] pipelines this communication in a ring topology, incrementally exchanging local key-value blocks and computing partial attentions at each stage. Above methods fundamentally rely on the costly communication of large $K, V$ matrices.

## 2.2 Attention Outputs Exhibit Low-Rank Structure

Our compression scheme is inspired by the observation that the attention outputs of pretrained transformers lie on a low-dimensional manifold. To support this, we analyze publicly available checkpoints of large-scale pretrained LLMs and examine their attention activations. Fig. 1 presents an illustration of LLAMA 70B.

Specifically, we measure the *stable rank* of the query ($Q$), key ($K$), and value ($V$) activations across each attention layer. The stable rank of a matrix $A \in \mathbb{R}^{n \times d}$ is defined as: $\mathrm{srank}(A) = \frac{\|A\|_F^2}{\|A\|_2^2}$, where $\|A\|_F$ denotes the Frobenius norm and $\|A\|_2$ the spectral norm. Unlike the conventional matrix rank – which is highly sensitive to small perturbations and numerical noise – the stable rank offers a robust, continuous measure of effective dimensionality. This makes it particularly suitable for characterizing learned neural representations, where numerous singular values are typically small yet non-zero due to noise or over-parameterization.

As depicted in Fig. 1, the stable ranks of attention activations remain low across all layers. Interestingly, $Q$ and $K$ generally exhibit slightly lower ranks than $V$, indicating a higher degree of compressibility [1]. This observation underpins our approach, leveraging low-rank factorization for efficient compression. Further evidence of this phenomenon in other architectures is provided in Appendix B. Next, we formalize this idea.

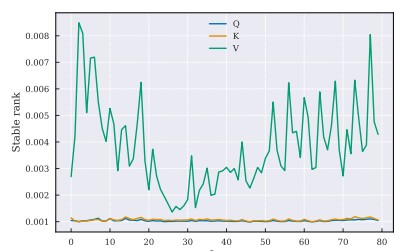

Figure 1: **Attention outputs of LlaMa-70B.** Shown is the empirical rank of the $Q$, $K$, and $V$ activations, normalised by their maximum possible rank, for every layer of the official LLAMA 70B checkpoint. All three projections are extremely low-rank: $Q$ and $K$ sit at roughly $0.1\%$ of full rank, while $V$ is slightly larger at $\sim 0.5\%$.

## 3 Method

We now present our proposed method for efficient context-parallel transformer training. First, we formalize how the empirically observed low-rank structure in attention activations enables effective compression. Next, we explain why using a fixed subspace for compression can be overly restrictive, motivating our joint learning strategy that adaptively optimizes both the projection subspace and the attention weights (§3.1). We then introduce a computationally efficient reparameterization approach that maintains optimality guarantees while significantly reducing overhead (§3.2). Finally, we describe how to reduce communication costs by dynamically compressing attention activations through per-chunk rotations and demonstrate how the model can seamlessly revert to a *standard transformer architecture* at inference time (§3.3–3.5).

In §. 2.2, we saw that the $Q$, $K$, $V$ activations of large pretrained transformers exhibit a pronounced low-rank structure (Fig. 1). This finding implies that it is feasible to transmit only the low-dimensional components of these activations between devices, thereby achieving near-lossless compression in practice. Formally, let the columns of an orthonormal matrix $U \in \mathbb{R}^{d \times r}$, with $r \ll d$, span the dominant subspace of the activations. Rather than communicating the full local activation matrix $Z = X^{(i)}W \in \mathbb{R}^{n_i \times d}$, where $Z \in \{K, V\}$ denotes key/value activations, $W \in \{W_k, W_v\}$ and $X$ is the input to the attention layer, we can transmit only its compressed representation: $Z_{\mathrm{comp}} = X^{(i)}WU \in \mathbb{R}^{n_i \times r}$. The original activations can then be reconstructed at the receiving node as: $Z \approx Z_{\mathrm{comp}}U^\top$. This compression method preserves all information within the subspace spanned by $U$, and is lossless when activations lie entirely in this subspace. Equivalently, this projection can be folded onto the attention weights and be interpreted as factorizing them into a low-rank representation: $W = B(UU^\top), \quad B \in \mathbb{R}^{d \times d}$.

**Sub-optimality of a fixed subspace.** The formulation above implicitly assumes that an *a priori* choice of $U$ is sufficiently expressive for every layer and every chunk in every optimization stage. It

---

[1]We only need to compress $K$ and $V$ since $Q$ can remain local.

is straightforward to see where this assumption can break down. Even if there is an optimal low-rank attention weight matrix, restricting weights to the form $W = BUU^\top$ limits the search to the column space of $U$. If this space does not contain the true optimum, the model may converge to a suboptimal solution. In short, fixing $U$ can prevent the model from reaching the best possible performance.

## 3.1  Joint Optimization over a Product Manifold

To address the limitations of a fixed $U$, we propose jointly optimizing factorization $W = BUU^\top$. Specifically, we simultaneously learn both the subspace representation $U$ and the matrix $B$ on the product manifold: $\mathcal{M} = \mathbb{R}^{d \times d} \times \mathrm{St}(n, r)$. Here, $B \in \mathbb{R}^{d \times d}$ is optimized in standard Euclidean space, whereas $U$ resides on the Stiefel manifold $\mathrm{St}(n, r)$, where updates can be naturally performed via Riemannian gradient descent [2]. The following result establishes that this joint optimization achieves linear convergence under gradient descent.

> **Convergence.** Let $\Phi(W, \vartheta)$ be a smooth loss and consider the factorization $W = BUU^\top$ for attention weights where $B \in \mathbb{R}^{d \times d}$, $U \in \mathrm{St}(d, r)$, and $\vartheta \in \mathbb{R}^p$ denotes all other parameters. Minimizing the reparameterized objective $\hat{\Phi}(B, U, \vartheta) = \Phi(BUU^\top, \vartheta)$ over the product manifold $\mathcal{M} := \mathbb{R}^{d \times d} \times \mathrm{St}(d, r) \times \mathbb{R}^p$ with Riemannian gradient descent, and under mild assumptions, yields *Q-linear* (geometric) convergence to a first-order stationary point. For the formal result and proof, see Lemma 1 (Appendix).

Note that since $\|U\| = 1$, the factorized objective remains Lipschitz smooth, and the convergence result naturally follows from the standard gradient descent theory on both Euclidean and Riemannian manifolds. We include a full proof in Appendix A for completeness, explicitly treating the product manifold structure and assuming a Polyak–Łojasiewicz (PL) condition.

## 3.2  Reducing Computational Cost via Reparameterization of $U$

Direct optimisation of $U$ on the Stiefel manifold $\mathrm{St}(n, r)$ via Riemannian gradient descent provides strong theoretical guarantees but is costly: after every Euclidean update, $U$ must be re-orthonormalised (the standard "retraction" on to the manifold), which is performed with a QR or SVD factorisation to restore $U^\top U = I_r$. To mitigate this, we use an efficient reparameterization of $U$ using a fixed orthonormal basis $\overline{U}$ and a learnable rotation $R(\theta) \in O(d)$:

$$U(\theta) = R(\theta)\,\overline{U},$$

where $O(d)$ denotes the orthogonal manifold consisting of all $d \times d$ orthonormal matrices. If the mapping $\theta \mapsto R(\theta)$ is sufficiently expressive, rotations $R(\theta)$ can fully parameterize $O(d)$, preserving the representational power of the manifold while significantly reducing computational overhead.

**Preservation of geometry and stationary points.**  Reparameterizing $U$ as $U(\theta) = R(\theta)\,\overline{U}$ moves the orthonormal constraint onto an unconstrained Euclidean variable $\theta$, eliminating expensive QR/SVD steps and letting us run ordinary SGD/Adam in $\theta$-space. *A natural concern is that this change of variables might distort the loss landscape and hinder optimization;* however, we show that this is not the case. The chain rule shows $\nabla_\theta \widehat{\Phi}(B, \theta, \vartheta) = D_\theta U(\theta)^\top \mathrm{grad}_U \Phi(B, U(\theta), \vartheta)$, so $\nabla_\theta \widehat{\Phi}$ is exactly the pull-back of the original Riemannian gradient. Thus, the first-order critical points remain unchanged. The following statement formalizes this result.

> **Equivalence of stationery points.** Under the reparameterization $U = R(\theta)\,\overline{U}$, minimizing $\hat{\Phi}$ possesses exactly the same local minima and strict saddle points as minimizing $\Phi$. For the formal result and proof, see Theorem 1 (Appendix).

Thus, we can effectively represent and optimize the projection subspace implicitly through rotations without compromising the quality or optimality of solutions.

---

[2]The Stiefel manifold $\mathrm{St}(d, r)$ is defined as the set of all $d \times r$ matrices with orthonormal columns, formally given by $\mathrm{St}(d, r) = \{U \in \mathbb{R}^{d \times r} : U^\top U = I_r\}$.

### 3.3 Reducing the Communication Cost

The reparameterization $U(\theta) = R(\theta)\,\overline{U}$ allows us to locally cache the fixed orthonormal frame $\overline{U}$ at each node, and transmit only the parameters $\theta$. However, to fully parameterize rotations in the orthogonal group $O(d)$, one typically requires $\frac{1}{2}d(d-1)$ parameters, *i.e.*, $\theta \in \mathbb{R}^{d(d-1)/2}$. We show next that in practice performing a dense search over all possible rotations is unnecessary. Specifically, we can obtain a trade-off between the search space and the communication efficiency by controlling the dimensionality of $\theta$.

To achieve a more compact representation for the communication cost reduction, we select a small set of fixed skew-symmetric matrices $\{A_1, \ldots, A_k\} \subset \mathfrak{o}(d), A_i^T = -A_i$ (where $\mathfrak{o}(d)$ denotes the Lie algebra of the orthogonal group) and define the corresponding $k$-dimensional Lie subgroup [16, 12]:

$$\mathcal{R}_k = \left\{ R(\theta) = \exp\left( \sum_{l=1}^{k} \theta(l) A_i \right) \mid \theta \in \mathbb{R}^k \right\},$$

where $\theta(l)$ is the $l^{th}$ element of $\theta$. Because the exponential map is a local diffeomorphism around $\theta = 0$, the set $U(\theta) = R(\theta)\,\overline{U}$ forms a $k$-dimensional submanifold of $\mathrm{St}(d, r)$ for sufficiently small $\|\theta\|$. Choosing $k \ll \frac{1}{2}d(d-1)$ thus provides a favorable trade-off between communication cost and representational flexibility. Importantly, our earlier result on the absence of spurious minima remains valid provided an optimal frame $U_\star$ lies within (or sufficiently close to) the reachable manifold $\{R\overline{U} : R \in \mathcal{R}_k\}$, as the mapping $\theta \mapsto U(\theta)$ remains locally surjective onto this manifold.

### 3.4 Dynamic mixtures of subspaces via per-chunk rotations

§ 3.3 depicted that the rotation dimension $k$ controls a trade-off between representational flexibility and communication efficiency. With a well-chosen a priori $\overline{U}$, it becomes feasible to use a small $k$, restricting the optimization to a local neighborhood around $\overline{U}$.

We generate this prior through a short, uncompressed **warm-up phase**, in which the model is trained for a small number of iterations ($< 500$) using a reduced context length to avoid communication bottlenecks. After this phase, each node computes the top $r$ principal components of its local attention weights and stores them as a fixed subspace basis $\overline{U} \in \mathrm{St}(d, r)$. Empirical evidence from prior work on weight–subspace stabilization (e.g., [13, 18]) suggests that dominant activation subspaces emerge early in training, supporting this strategy.

**Per-sample adaptation.** Using a single global rotation for all inputs may underfit heterogeneous data. To retain expressivity *without increasing* $k$, we introduce a lightweight mechanism to predict a unique rotation parameter $\theta$ for each sequence chunk. Recall that, in context-parallel training, each node $i$ processes a distinct chunk $X_i \in \mathbb{R}^{n_i \times d}$ from the input sequence. For an attention output chunk $Z_i = W X_i$, we employ a small linear prediction head: $\psi : \mathbb{R}^d \to \mathbb{R}^k, \quad \theta = \psi\left(Z_{\mathrm{avg}, i}\right)$, where $Z_{\mathrm{avg}, i}$ is the average attention output of the chunk, generating chunk-specific rotation parameters. Given a set of preshared skew-symmetric generators $\{A_l\}_{l=1}^{k} \subset \mathfrak{o}(d)$ cached locally on each node, we construct the rotation as: $R(\theta_i) = \exp\left( \sum_{l=1}^{k} \theta_i(l) A_l \right) \in \mathcal{R}_k \subset O(d)$. Locally, keys and values are compressed as $Z_{\mathrm{comp}, i} = Z_i\, R(\theta_i)\,\overline{U} \in \mathbb{R}^{n \times r}$. $(Z_{(\mathrm{comp}, i)}, \theta_i)$ is then broadcasted, and the receiving nodes reconstructs the keys/values as: $Z_i \approx Z_{(i, \mathrm{comp})}\,\overline{U}^{\top} R(\theta_i)^{\top}$. Note that peak memory is dominated by the attention computation, scaling as $\mathcal{O}(n_i^2)$, making the linear head's overhead $\mathcal{O}(dk)$ negligible—an observation we also demonstrate empirically. The overall procedure is summarized in Algorithm 1.

**Bandwidth cost.** In our method, each node transmits $nr$ floats (activations) in $\tilde{Z}$ and $k$ additional scalars in $\theta$. Typically, we have $k \ll nr \ll nd$, ensuring low communication overhead. Remarkably, we found that even using $k = 1$ – a single rotation angle that defines a plane – is sufficient to preserve training stability and input-adaptive flexibility, achieving bandwidth efficiency comparable to that of a fixed global rotation.

In implementation, we set $S \sim \mathcal{N}(0, 1)^{d \times d}$ to be fixed and define the skew-symmetric generator $A := \frac{\theta}{\|S - S^\top\|_{\mathrm{F}}} (S - S^\top), \quad A^\top = -A, \quad \theta \geq 0$. For $\theta \in \mathbb{R}$ we set the rotation $R(\theta) = \exp(\theta A) \in O(d)$, so $A$ fixes the rotation plane while $\theta$ sets its magnitude.

---

**Algorithm 1** Compression-aware context parallel attention (per node, per head)

---

**Require:** Input $X \in \mathbb{R}^{n_i \times d}$, Attention weight $W \in \mathbb{R}^{d \times d}$, Warm-started basis $\bar{U} \in \mathbb{R}^{d \times r}$, learnable linear head $\psi : \mathbb{R}^d \to \mathbb{R}^m$, sync interval $c$, current step $t$

1: Compute local keys and values: $Z \leftarrow XW$
2: $Z_{\text{avg}} \leftarrow \texttt{MeanToken}(Z)$
3: $\theta \leftarrow \psi(Z_{\text{avg}})$                                                  ▷ Compute rotation param from local chunk
4: $U \leftarrow R(\theta)\bar{U}$                                                  ▷ Construct data-dependent subspace
5: Compress: $Z_{\text{comp}} \leftarrow ZU$
6: Broadcast $(Z_{\text{comp}}, \theta)$ to all other nodes
7: Receive $(Z_{(\text{comp},j)}, \theta_j)$ from all other nodes $j$
8: **for all** received $(Z_{(\text{comp},j)}, \theta_j)$ **do**
9:     $U_j \leftarrow R(\theta_j)\bar{U}$
10:    $Z_j \leftarrow Z_{(\text{comp},j)} U_j^\top$                                             ▷ Decompress
11: **end for**
12: Aggregate global $Z_g \in \{K_j, V_j\}, \forall j$ from all nodes
13: Compute blockwise attention: $A \leftarrow \texttt{Softmax}(QK^\top/\sqrt{d})V$
14: **if** $t \bmod c = 0$ **then**
15:    $W \leftarrow \texttt{AllReduceAvg}(W)$                                ▷ Sparse sync of attention weights
16: **end if**

---

**Second-order approximation.** Since $A$ is skew-symmetric, its spectral norm satisfies $\|A\|_2 = \theta$. For sufficiently small $|\theta| \leq \epsilon \ll 1$, the rotation matrix $R(\theta)$ admits a second-order Taylor approximation:

$$R(\theta) \approx I + \theta A + \tfrac{1}{2}\theta^2 A^2. \tag{1}$$

This approximation provides two key advantages. 1) **Computational cost:** scaling as $\mathcal{O}(d^2)$, in contrast to the exact matrix exponential computation (e.g., by Padé or Schur decomposition), which scales as $\mathcal{O}(d^3)$. 2) **Near identity bias:** it induces a beneficial near-identity bias, effectively acting as an approximately unbiased estimator of identity $I$ when $\theta$ is small and centered around zero (enforced via clipping). In this regime, higher-order terms vanish in expectation, yielding $\mathbb{E}[R(\theta)] \approx I$. This property allows rotations to remain close to the initial warm-start subspace $\bar{U}$, facilitating controlled local adaptation without significant drift. By fixing $A$ and using a scalar $\theta$, we achieve a communication complexity of $\mathcal{O}(nr)$, significantly lower than the naive $\mathcal{O}(nd)$.

Attention weights must still be synchronized across devices, but they evolve far more slowly than activations [6, 5]. We therefore average the corresponding weights only every $c$ steps; in all experiments we use $c = 200$, which incurs negligible communication overhead.

## 3.5 Unplugging the Projection Components

Our method augments the transformer architecture with two non-standard components: (i) a small linear *rotation head* predicting the rotation parameters $\theta$, and (ii) low-rank *projection matrices* $U$ used for compressing activations. Although these components pose minimal overhead during training, strict API compatibility with off-the-shelf transformer models might be necessary for certain downstream applications.

As training proceeds, the learnable weights associated with our auxiliary projection heads collapse onto the data–dependent subspaces they steer. Once the model is close to convergence we can therefore *drop these heads entirely*, reverting to a vanilla Transformer without losing the predictive gains accumulated during training. The following result formalizes the collapse mechanism.

> **Bound on "idle" attention directions with data dependent projectors.** Let the sample projector be $P(x) = U(x)U(x)^\top$. Pick any other projector $Q$ that projects onto an arbitrary subspace. Define the average overlap $p_Q := \mathbb{E}_x\big[\|P(x)Q\|_2\big] \in [0,1]$. Run stochastic gradient descent with weight decay $\lambda > 0$. Then, the attention weights that lie inside the $Q$-subspace obey $\lim_{t\to\infty}\big\|W^{(t)}Q\big\|_F \leq \frac{p_Q L}{\lambda}$ for an $L$ Lipshchitz bounded loss. Hence, if the data almost never excites those directions ($p_Q \ll 1$), the corresponding weights shrink away. That is, idle directions are pruned for free. For the formal theorem and proof, see Theorem 3 (Appendix).

Once the weights have collapsed onto their data–aligned subspaces, both the rotation head and its basis matrix $U$ are redundant. We can therefore *detach* these components and perform a brief,

low-learning-rate fine-tuning pass to polish the remaining parameters. As Fig. 4 shows, the loss curve remains smooth across this transition, indicating that no *optimisation shock* is introduced.

At inference time the model is now *indistinguishable* from a standard transformer: it adds **zero** extra parameters, requires no custom kernels, and is fully compatible with existing deployment pipelines.

## 4 Related Work

**Decentralized training.** Decentralized learning dispenses with a central coordinator, instead relying on a collective of *autonomous* devices that cooperate over mesh-style networks to train large-scale models. These devices are typically heterogeneous and geographically dispersed, confronting links of nonuniform latency and bandwidth. The foundational theory on convergence and robustness has been established by [24, 22, 21], while complementary systems work has demonstrated practical viability on real clusters [41, 7]. Most prior art, however, is confined to DDP settings [24, 22, 21, 7], limiting model size to the aggregate memory of an individual node. Note that this is a comparatively well studied domain, and is orthogonal to the unexplored decentralized context parallel setting that we explore. A notable work in DDP domain is Power Gossip [48], which replaces synchronous all-to-all communication with gossip-style information exchange among neighboring replicas arranged in a mesh. Its key insight is that, when each replica trains independently via local SGD, the pairwise weight differences evolve in a low-rank sub-space, enabling them to efficiently compress the weight differences during gossip. Another interesting DDP method is Photon [42], where its communication savings stem primarily from infrequent gradient exchanges rather than from any explicit compression scheme. Such skip-sync approaches are infeasible in context-parallel pipelines, where activations must be transferred between nodes at every forward and backward pass. Nevertheless, these DDP-style techniques are orthogonal to our method and could be combined with it in hybrid setups.

Scheduling-oriented approaches such as SWARM parallelism [39] and Tasklets [53] alleviate straggler effects and network stochasticity, yet they still inherit the communication overhead intrinsic to the decentralized setting. In contrast, we introduce the first communication-compression strategy tailored to CP, removing a critical bandwidth bottleneck that has thus far hindered scaling decentralized models across larger context windows.

**Context-parallel attention.** For single-device long-sequence processing, sparse approximations such as BigBird [54] cut attention complexity to $O(n)$, while IO-aware exact kernels like FlashAttention [33] maximize hardware throughput with tiling and on-chip caching. Recent systems research parallelizes the *sequence dimension* itself [25, 37, 15]: Blockwise Parallel Transformers overlap compute and ring-all-reduce to achieve near-linear speedups on sequences of 32K tokens [28], and RingAttention extends the idea to virtually unlimited contexts via pipelined block exchanges [29]. These methods, however, still broadcast full key/value tensors. Our approach instead transmits a compact low-rank representation plus a lightweight rotation, reducing bandwidth while preserving exact attention semantics and thus complementing existing context-parallel frameworks.

## 5 Experiments

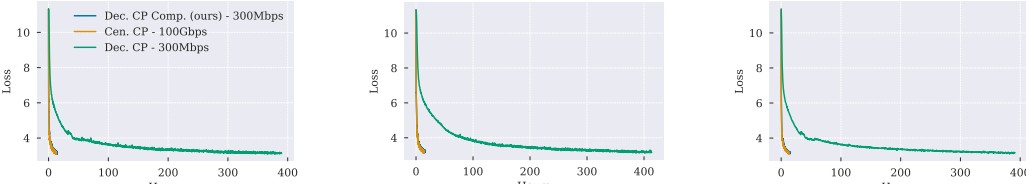

Figure 2: **Convergence in low-bandwidth settings.** From left to right: Fineweb, C4, and BookCorpus. The training curves are presented against wall-clock time for an 8-layer (800M) model trained with a 132K context window parallelized across 8 GPUs. Decentralized models utilize 300Mbps connections while the centralized model has datacenter-grade 100Gbps links. Our compressed model achieves on-par convergence to the centralized model, even under a 300Mbps bandwidth budget. In contrast, the non-compressed decentralized model with 300Mbps links suffers from significantly slower convergence.

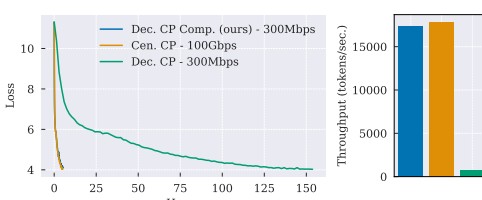

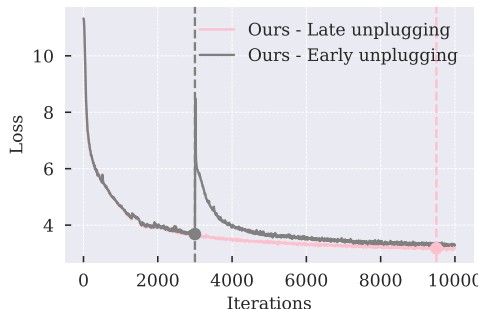

Figure 3: **Scaling across parallelism strategies.** Our compression based CP scheme can be seamlessly fused with other parallel training strategies. We train a 3B-parameter model (32 layers) with both pipeline parallel and CP enabled across 32 A100s. Our compressed approach yields substantial throughput gains over uncompressed decentralized CP and nearly matches the performance of centralized CP.

Figure 4: **Unplugging projection components.** After sufficient training, the rotation head and projection layers can be removed—reverting the network to a vanilla transformer—without impairing convergence. Training curves with dashed lines marking projection removal points. Late removal preserves convergence (see §. 3.5), while early removal causes a temporary disruption followed by surprisingly rapid recovery.

## 5.1 Experimental Setup

We evaluate decoder-only models on three large-scale corpora – FineWeb (FW) [30], C4 [35], and BookCorpus (BC) [56]. For each dataset, we reserve $10\%$ of the training split for validation. All model backbones follow LLAMA 3 [11]; exact model specifications are given in the corresponding sections. We use a `base-learning-rate` $= 3 \times 10^{-4}$ with linear warm-up and decay, and apply a `weight-decay` $= 0.01$. Every transformer layer is compressed except for the final block, where $K$ and $V$ projections are compressed by $98\%$ and $95\%$ (overall $96.5\%$), respectively by choosing $r$ w.r.t. $d$ appropriately. We use the GPT2 tokenizer for all models.

## 5.2 Bandwidth efficiency in decentralized settings

We train an 8-layer, 800M-parameter model (`embedding-size = 2048`, `attention-heads = 8`) under two network settings: a centralized 100Gbps fabric and decentralized 300Mbps internet-grade links. Using CP, we process a sequence length of 132K tokens across eight A100 GPUs connected at the respective bandwidths. Fig. 2 shows that vanilla CP over a 300Mbps link is more than $20\times$ slower compared to a centralized 100Gbps mesh. With our compression, the same 300Mbps setup converges almost as fast as the centralized baseline.

Table 1: **Design ablations** (val. perplexity ↓). All models are trained for 10K steps with a 132K context. Second-order approximations preserve performance, while overcompressing $V$ degrades it.

| SETTING | FW | C4 | BC |
|---|---|---|---|
| **Ours** | **22.64** | **23.33** | **25.27** |
| Ours + Fixed $\bar{U}$ | 26.57 | 27.11 | 30.33 |
| Ours + Rand. rot. $R(\theta)$ | 24.93 | 25.17 | 29.58 |
| Ours - 2nd-order approx. | **22.64** | **23.33** | **25.27** |
| Ours - No warm start | 26.63 | 26.91 | 30.15 |
| Ours $(K, V \to 98\%)$ | 24.74 | 24.99 | 29.46 |
| Ours $(K \to 99\%, V \to 95\%)$ | 24.68 | 24.91 | 29.22 |

**Validation.** Table 2 reports test-time performance of the trained models. To this end, we train each model up to its compute-optimal point, following the Chinchilla scaling law [17]. Specifically, for our 800M-parameter models, we use a $1 : 20$ model-to-token ratio and train for 16B tokens on each dataset. Remarkably, our compressed decentralized model matches, and even slightly outperforms, the perplexity of the centralized model at the same number of training iterations, while delivering significantly higher throughput than vanilla (uncompressed) CP over commodity links. Training the uncompressed model to completion over low-bandwidth links is computationally infeasible (estimated at over 150 days), so we report only its throughput (TPS) in this setting.

Table 2: **Validation perplexity ($\downarrow$) and throughput (TPS).** All models are trained with a 132K context window to the compute-optimal point [17] (16B training tokens). Our method yields a $20\times$ TPS boost while slightly outperforming centralized CP in perplexity, with minimal memory overhead.

| MODEL | FW | C4 | BC | TPS | MEM (GB)/GPU |
|---|---|---|---|---|---|
| Cen. CP - 100Gbps | 17.18 | 17.51 | 17.88 | 56K | 38.4 |
| Dec. CP - 300Mbps[†] | – | – | – | 2.7K | 38.4 |
| **Dec. CP Comp. - 300Mbps (ours)** | **17.06** | **17.47** | **17.81** | **55K** ($\times 20$) | 38.7 (+0.7%) |

[†] Training uncompressed models to convergence at 300Mbps is infeasible ($>150$ days); only throughput is reported.

## 5.3 Ablations

We perform ablations on 800M-parameter models with a 132K context across eight A100 GPUs (see Table 1). Models using learned rotations outperform those with fixed or random projections. The second-order exponential approximation does not impact performance, confirming its adequacy. Omitting the warm-start initialization of principal directions ($\bar{U}$) noticeably degrades results, highlighting the importance of this prior.

**Scaling:** Our compression based CP scales well and can be seamlessly fusing with other parallel training strategies. We scale the model to 32 layers (3B parameters) with both pipeline parallel and CP enabled over 32 A100s (Fig. 3) and achieve a significant throughput gain.

**Reparameterization:** A key step of our method is reparameterizing $U$ which bypasses expensive Riemannian operations (QR/SVD pullbacks). As shown in Table 4, this reparameterization significantly improves throughput (TPS). More ablations against architecture choices are provided in Appendix C.

Table 3: **Effect of warmup steps** (val. perplexity $\downarrow$). All models are trained for 10K steps with a 132K context. The method is not highly sensitive to the number of warmup steps.

| WARMUP-STEPS | PERPLEXITY |
|---|---|
| 0 | 26.63 |
| 100 | 24.44 |
| 300 | 22.66 |
| 500 | 22.64 |
| 1000 | 22.87 |
| 2000 | 22.64 |
| 5000 | 22.71 |

**Warmup steps:** To measure the effect warmup steps of we conducted an ablation study varying the warm-up duration and evaluated the resulting perplexity on the FineWeb dataset. The results are shown in Table 3. As demonstrated, even with a reduced warm-up of 300 steps, the model achieves comparable performance, indicating no significant degradation. In practice, we default to 500 steps to provide a safe and stable baseline. This study further emphasizes the lightweight and robust nature of our warm-up strategy, especially in contrast to the more elaborate scheduling mechanisms commonly employed in modern LLM pre-training. Note that the perplexity differences are minor and stable, indicating performance is stable after 300 warm-up steps.

## 5.4 Unplugging Projections and Rotation Heads

As discussed in §. 3.5, practitioners may prefer reverting to a standard transformer after pretraining for compatibility with downstream frameworks. We empirically validate our theoretical prediction that attention weights progressively align with the projection subspace, allowing safe removal of projection layers and rotation heads near the end of training. Fig. 4 shows that removing these components late preserves convergence, while doing so prematurely disrupts training.

## 5.5 Comparison Against Baselines

As no prior baselines exist for CP compression, we construct two: (i) **Sparsification**—a Top-10% scheme (90% compression), transmitting only the largest-magnitude entries of the $K, V$ chunks, inspired by common DDP compression methods; (ii) **Quantization**—a 4-bit quantization (75% compression) of the $K, V$ activations prior to transmission, following standard practices in activation

compression. As shown in Fig. 5 (left), we outperform these baselines comprehensively (132k context window) even when using a more aggressive compression rate of $96.5\%$.

For completeness, we also compare against long-context models BigBird [54] and CosFormer [34], which are not designed for CP and can handle at most 32K tokens on an A100. For a fair comparison, we apply our compression to CP across four GPUs, each processing 8K tokens. As shown in Fig. 5 (right), both baselines exhibit significantly worse convergence than our method. All experiments are performed on 800M parameter models.

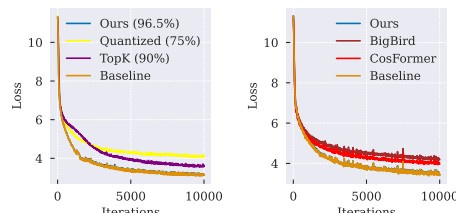

Figure 5: **Baseline comparisons.** *Left:* Because no method currently compresses context-parallel training, we build two baselines; Top-$k$ sparsification and quantization. *Right:* We also compare with long-context models BigBird and CosFormer. Both are limited to 32K tokens on A100 GPUs, so all models are evaluated at that length. In both panels, our compressed CP curve is nearly indistinguishable from the uncompressed reference, whereas every baseline falls well short.

## 6 Conclusion

We propose the first compression method that enables context-parallel training of language models in decentralized environments with low-bandwidth interconnects. Our approach supports training with context lengths over 100K tokens on isolated GPUs connected via internet-grade links (e.g., 300Mbps), while matching the wall-clock convergence of centralized systems with high-speed (100Gbps) connections. Additionally, our method preserves compatibility with standard transformer architectures by allowing the projection layers to be removed after training, facilitating seamless deployment in downstream frameworks. We provide a theoretical analysis of the key properties of our method and validate its effectiveness through an extensive empirical evaluation.

## 7 Limitations

Our compression method delivers near-lossless convergence in context-parallel training, but several open questions remain. First, alternative reparameterisations beyond simple subspace rotations may unlock further accuracy or efficiency gains. Second, the method's surprising ability to locate good minima even as the search space is heavily reduced (via very low-dimensional $\theta$) lacks a rigorous explanation; its ties to recent work on implicit regularisation and lottery-ticket-style phenomena deserve closer study. Despite these gaps, this work establishes the first baseline for context-parallel compression and we hope it spurs deeper theoretical and empirical exploration.

Table 4: **TPS gain from design choices.** Reparameterization and second-order approximation yield significant throughput improvements.

| SETTING | TPS ($\uparrow$) |
|---|---|
| **Ours** | **55K** |
| w/o reparam. | 37K |
| w/o $2^{nd}$ ord. approx. | 30K |

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
