# OpenReview forum: "Mixtures of Subspaces for Bandwidth Efficient Context Parallel Training"
_NeurIPS.cc/2025/Conference — NeurIPS 2025 poster_

### Official Review · Reviewer_Pnue · 2025-06-23

**Clarity:** 3
**Significance:** 4
**Originality:** 4
**Rating:** 5
**Confidence:** 3

**Summary:**

This paper addresses the bandwidth bottleneck of context-parallel (CP) training for very-long-context transformers. CP divides a single long sequence across GPUs; every layer therefore requires an all-gather of full key/value (K,V) activations, which is trivial on datacentre interconnects but prohibitive on commodity networks. Similar to previous work, the authors show empirically that K,V lie on an extremely low-rank manifold (0.1-0.5% stable rank). They propose Mixtures-of-Subspaces (MoS), where each attention weight is factorised so that its outputs are constrained to an r-dimensional subspace.
A short warm-up run provides an initial orthonormal basis for the low rank subspace $\bar U$, and thereafter the model learns only a rotation $R(\theta)$ of that basis together with a low-rank matrix $B$. This re-parameterisation keeps the columns of $U=R(\theta)\bar U $ orthonormal without per-step QR/SVD. Projecting $K,V$ onto this subspace reduces communication from $O(nd)$ to $O(nr)$.

On eight A100 GPUs linked by 300 Mbs Ethernet, MoS reduces CP traffic by ≈20x and matches the wall-clock convergence of an uncompressed baseline on 100 Gb s InfiniBand using an 800 M-parameter, 132 K-token model. Because weight-decay shrinks directions outside the learned subspace, the projection and rotation heads can be “unplugged” after training. Hereby, the final model is a standard transformer.

**Questions:**

1. Warm-start sensitivity:  How many uncompressed steps are needed to build \bar U? Does the answer change with dataset choice?
2. Unplugging: How sensitive is the final PPL to the number of fine-tune steps and weight-decay value used when removing the projector? I see Figure 4 shows an example of this, but how sensitive is it between the late and early “unplugging” steps in this figure?
3. Training-loss vs validation gap: In Figs 2–3 the “cen-CP 100 Gbps” curve appears to reach lower training loss than MoS, yet Table 1 reports better validation perplexity for MoS. Can you comment on this discrepancy? A zoom-in plot (similar to App.\ Fig 14) in the main paper would help.
4. Variance and robustness: All main curves and ablations are single runs. Could you provide some error bars for some of the smaller ablation experiments?
5. Relation to DeepSeek MLA: DeepSeek’s Mixed-Linear Attention (MLA) also projects K and V to a fixed low dimension, but as I understand it, it keeps the keys/values low-rank all the way through the attention kernel, giving a lasting memory/compute win, whereas MoS re-expands the compressed blocks after the all-gather and therefore targets communication rather than peak RAM.
Could you clarify this distinction and comment on whether there is any practical benefit (or incompatibility) in combining MoS with MLA

**Ethical Concerns:**

["NO or VERY MINOR ethics concerns only"]

**Final Justification:**

The authors provided a clear rebuttal that addressed my concerns, and my questions are now resolved. The authors showed insensitivity to the warm-start duration, provided a simple, relatively robust “unplugging” heuristic.
I find the results impressive and the method appealing, and I see it as a practical approach for bandwidth-limited context-parallel training. I’ll keep my score at 5 (accept), noting the setting is fairly specific to GPUs without high-speed interconnects.

**Limitations:**

Yes, Section 7 discusses theory gaps and the unexplained success of very low-dimensional rotations.

**Paper Formatting Concerns:**

No concerns.

**Quality:**

4

**Strengths And Weaknesses:**

**Quality**
Solid emprical results: Three data sets, 132 K-token context, and a ~20 × wall-clock speed-up on commodity 300 Mbs links. Interesting ablations: Layer-wise rank study, rotation-size sweep, comparisons to 4-bit and Top-10 % sparsity. Theory: Proof that the re-parameterised optimisation retains convergence guarantees.
No variance/error bars. Table 2 and Fig. 5 would be more convincing with ± 1 σ over at least three seeds. (I understand this is harder (costly) for the full runs in Table 1.)

**Clarity**
Clear high-level motivation; algorithms and all hyper-parameters are spelled out; code is included in supplementary.
Figures 2 and 3 are hard to read: the overlapping “cen-CP 100 Gbps” vs “dec-CP 300 Mbps (comp)” curves look identical but the axes are tiny. A zoom-in would be useful in the main paper. More detailed on explored hparams could also make it easier to reproduce experiments.

**Significance**
First method that makes Context paralism practical on commodity networks, potentially opening > 100 K-token training to a far wider pool of researchers. Orthogonal to existing attention kernels since it does not require architectural changes at inference.
Impact hinges on the low-rank property holding for other modalities or tasks; scalability beyond the 800 M-parameter (and 3B) range is not demonstrated.

**Originality**
The paper is the first to use low-rank structure specifically to minimise all-gather traffic in context-parallel training. This communication-first application of low-rank projections plus the “unplug-after-training” trick that removes the projector at inference is, to my knowledge, new and distinguishes the work from prior memory- or weight-centric compression schemes.

---

> ### Author Rebuttal · Authors · 2025-07-31
>
> We appreciate the reviewer’s positive remarks, describing our work as addressing “a highly relevant and timely problem,” providing “solid empirical results,” delivering “a clever technical approach,” and representing “the first practical method for context parallelism.” Below, we respond to each of the reviewer’s specific concerns in turn.
>
> ## 1) Warm-start sensitivity: How many uncompressed steps are needed to build \bar U? Does the answer change with dataset choice?
>
> We thank the reviewer for the thoughtful question. Below, we clarify that our warm-up strategy is **hyper-parameter agnostic** and fits naturally into the **context-length warm-up practices** widely adopted in modern LLM training.
>
> ---
>
> ### • Fixed warm-up configuration across all settings
> We employ the **same warm-up phase (500 steps with a 2 K context length) in every experiment across all the datasets**, regardless of
>
> * **Model size:** 800 M → 3 B parameters
> * **Depth:** 8 → 32 layers
> * **Target context:** 132 K (main paper) and 256 K (Fig. 13, supp.)
>
> This covers all major results (e.g., Fig. 3, 10 for 3 B, Fig. 12 for 2.5 B). We apply the identical warm-up to matched uncompressed baselines to ensure a fair comparison. Across these diverse setups we saw **no meaningful sensitivity** to the warm-up duration; the fixed schedule worked consistently well.
>
> ---
>
> ### • Observed insensitivity to warm-up length
> To further address the reviewer’s concern, we varied the warm-up duration and measured perplexity on **FineWeb**. We observed that the model’s performance remains stable even at **300 warm-up steps**, however,  we retain 500 steps as a conservative default. Note that Perplexity is computed as the exponential of validation loss; hence decimal-level changes are minor  (mostly the third decimal point in the loss).
>
> | Warm-up steps | Perplexity |
> |--------------:|-----------:|
> | –             | 26.63 |
> | 100           | 24.44 |
> | 300       | 22.66 |
> | 500 (default) | 22.64 |
> | 1 000         | 22.87 |
> | 2 000         | 22.64 |
> | 5 000         | 22.71 |
>
> These results confirm the **lightweight and robust** nature of our warm-up, especially relative to the more elaborate schedules (learning-rate, batch-size, context-length) common in large-scale LLM training.
>
> ---
>
> ### • Context-length warm-up is standard in frontier models
> Our approach aligns and fits naturally into the established practice of context length warm-up in state-of-the-art models:
>
> * **Llama-3 400B:** six-stage increase from 8 K → 128 K tokens&nbsp;[1]
> * **Qwen-2:** gradual growth from 4 K → 256 K tokens&nbsp;[2]
> * **Hugging Face SmolLM-3:** 4 K → 32 K → 64 K tokens&nbsp;[3]
> * **Li et al. (2022):** sequence-length warm-up lowers gradient variance, enabling GPT-3 to train with **8 ×** larger batches and **40 ×** higher learning rates&nbsp;[4]
>
> Our strategy of brief training at a shorter context, computing $\bar{U}$, then switching to full length, follows the same principle yet is **far lighter**: only **500 steps** at reduced length vs. thousands of iterations in the examples above.
>
> ---
>
> Modern LLM pre-training already combines warm-ups for **learning rate**, **batch size**, and **context length**, each with a far bigger impact on convergence. Compared with these, our method is **simpler** and **less sensitive** to tuning, while maintaining strong empirical performance.
>
> ## 2) Unplugging: How sensitive is the final PPL to the number of fine-tune steps and weight-decay value used when removing the projector?
>
> Thank you for the question. In our experiments, **removing the projection components during the last 30 % – 5 % of training consistently yields stable performance** across model sizes and datasets. Based on this observation, we recommend unplugging in that interval as a simple, effective heuristic. Notably, we see **no meaningful variance in perplexity** within this range, highlighting the robustness of the method.
>
> Below is an ablation on **FineWeb** (10 k total iterations). Perplexity is reported as $\exp(\text{validation loss})$, hence, decimal-level changes reflect very small differences (mostly the third decimal point in the loss). Even with an earlier removal (e.g., 5 k steps), the model recovers surprisingly quickly (as we mention in the paper), so the unplugging point is *not* a dominant factor in final performance.
>
> | Unplugging Step | Perplexity |
> |-----------------|-----------:|
> | **Not unplugged** | 22.64 |
> | 5 k             | 22.69 |
> | 7 k             | 22.64 |
> | 8 k             | 22.71 |
> | 9 k             | 22.71 |
> | 9.5 k           | 22.77 |
> |9.9 k  | 23.12
>
> These results confirm that unplugging anywhere in the final 30 - 5% of training maintains essentially a similar perplexity, making the procedure both **robust and easy to automate**. We use the same 0.1 weight decay throughout the training process.
>
>
> ## 3) Training-loss vs validation gap: In Figs 2–3 the “cen-CP 100 Gbps” curve appears to reach lower training loss than MoS, yet Table 1 reports better validation perplexity for MoS.
>
> Thank you for the keen observation. Table 1 reports scores **after each model is trained to its compute-optimal point**. For the **uncompressed centralized baseline**, reaching that point would require roughly **150 days**, which was beyond our experimental budget.  Therefore, in **Figs. 2–3** we instead plotted each run **only until the the curves first align to a reasonable training loss**, to illustrate how quickly each method descends. Because the centralized curve stops earlier, it has not yet entered the long tail of optimization.
>
> In the **final phase of training** (omitted from the figure), the **decentralized compressed model continues improving and ultimately slightly surpasses the centralized baseline**, which is why MoS achieves a better validation perplexity in Table 1. We will clarify this in the revised manuscript.
>
> ## 4) A zoom-in plot (similar to App.\ Fig 14) in the main paper would help.
>
> We will add this. Thank you.
>
> Certainly. We reran the **warm-up vs. no-warm-up** experiment with **three independent seeds** and report the mean and standard deviation (σ) of the resulting perplexities on **FineWeb**.  As perplexity is $\exp(\text{validation loss})$, the observed variation corresponds to changes mostly only in the third decimal place of the loss.
>
> | Method        | Run #1 | Run #2 | Run #3 | Mean ± σ |
> |---------------|-------:|-------:|-------:|----------|
> | w/ warm-up    | 22.64  | 22.59  | 22.66  | **22.63 ± 0.03** |
> | No warm-up    | 26.63  | 26.69  | 25.57  | **26.30 ± 0.63** |
>
> We will strive our best to provide similar multi-seed error bars for the remaining ablations in the revised version.
>
> ## 5) Relation to DeepSeek MLA
>
> We thank the reviewer for the insightful question. The reviewer’s understanding is correct: **MLA** projects **Q, K, V** to a low-dimensional latent space and performs attention computation *entirely* in that space. The primary goal is **memory and compute savings**, particularly for **KV caching during inference**. This complements our work in spirit. Both are motivated by the empirical observation that **QKV spaces are low-rank**, however, there are fundamental differences in their design goals and applicability: **MLA** reduces the dimensionality for attention *computation*, but still requires **each GPU to hold the full attention state** for its batch. **MoS** re-expands K and V and compute attention *block-wise* after the all-gather, which avoids increasing peak memory. It also **significantly reduces inter-GPU communication**.
>
> Adapting MLA directly to context-parallel training poses a **critical synchronization issue**: Since **Q is projected locally** on each GPU, the same query token (replicated across GPUs) would produce **inconsistent projections** unless the projection matrix is fully synchronized. This means that **identical Q tokens would produce different attention scores** when matched with the same K,V, impacting the attention scores. Synchronizing the Q projection matrix across GPUs (e.g., via weight sharing or frequent broadcast) introduces **heavy communication overhead**, defeating the goal of bandwidth-efficient training.  In contrast, MoS compresses and transmits K and V, and each receiving GPU knows the shared basis U and the low-rank parameters  $\theta$. Thus, U and $\theta$ are known to all GPUs, allowing consistent reconstruction of K and V after all-gather. This ensures that identical Qs yield consistent attention scores, regardless of which GPU computed them, maintaining semantic consistency across devices without extra communication. Therefore, while MLA and MoS are conceptually related in exploiting low-rank structure, their goals differ (memory vs. communication), and MLA is not directly compatible with context parallelism due to projection consistency issues. Nevertheless, MLA reinforces our empirical thesis, that QKV spaces are compressible without loss, and may inspire future designs that combine memory and communication efficiency in a unified framework.

---

> > ### Comment · Reviewer_Pnue · 2025-08-01
> >
> > Thank you for the clear rebuttal and added details. This addresses my questions well. I’ll keep my score at 5. I still see this as an interesting and practical approach for bandwidth-limited CP training.

---

### Official Review · Reviewer_WpGw · 2025-07-02

**Clarity:** 3
**Significance:** 1
**Originality:** 3
**Rating:** 3
**Confidence:** 4

**Summary:**

This paper introduces a novel method for bandwidth-efficient context-parallel (CP) training in decentralized environments. The authors propose dynamically compressing key and value activations using learned low-rank projections into mixtures of subspaces. By jointly optimizing both the subspaces and attention weights on a product manifold, and using per-chunk rotations with compact parameterization, the method achieves over 95% communication compression. The compressed activations can be transmitted efficiently even over low-bandwidth (e.g., 300Mbps) links, and decompressed without degrading model quality. Extensive experiments on long-context training (132K tokens) across datasets show that the proposed method achieves convergence comparable to centralized setups, with 20× speedup over uncompressed CP on commodity links.

**Questions:**

Please the the weakness part.

**Ethical Concerns:**

["NO or VERY MINOR ethics concerns only"]

**Limitations:**

yes

**Quality:**

2

**Strengths And Weaknesses:**

Strength:
1. The use of dynamic mixtures of low-rank subspaces and efficient rotation-based reparameterization is both novel and theoretically grounded.
2.  The paper includes careful breakdowns of design choices, including the role of warm starts, second-order approximations, and reparameterization.

Weakness:
1. The motivation is not persuasive. Why do we need decentralized long-context training over low-bandwidth connections? Why a long context is distributed across "internet-grade" linked devices if it has inherent long-range dependencies?
2. Context parallel introduces frequent communication operations. Pipeline parallel might seem to be a better choice for low-bandwidth training with a much smaller number of communication volumes and frequencies.
3. The surprising effectiveness of extremely low-dimensional rotations is acknowledged but not fully understood. The paper lacks comparison with other attention sparsification methods.

---

> ### Author Rebuttal · Authors · 2025-07-30
>
> We thank the reviewer for the constructive feedback and intriguing questions. Please find our answers below.
>
> ## The motivation is not persuasive. Why do we need decentralized long-context training over low-bandwidth connections? Why would a long context be distributed across "internet-grade" linked devices if it has inherent long-range dependencies?
>
> Thank you for raising this important question. We would like to clarify the motivation behind our setting and why decentralized long-context training is both **necessary** and **increasingly relevant**.
>
> ---
>
> ### 1. Centralized large-scale training is prohibitively expensive
>
> Modern LLMs are trained on massive HPC clusters with high-bandwidth interconnects (e.g., InfiniBand), which are extremely costly to build and operate. These clusters are typically accessible only to a few well-funded organizations. As a result, most researchers cannot afford to run large-scale experiments needed to rigorously evaluate new ideas. This creates a significant **barrier to entry** and limits scientific progress.
>
> ---
>
> ### 2. Volunteer computing offers a scalable and democratized alternative
>
> A promising alternative is **scaling out** via **volunteer computing**, where thousands of regular PCs contributed by volunteers provide compute. This paradigm has already been successfully applied in domains like language modeling [1], biology [2], high-energy physics [3], and astronomy. The aggregate FLOP capacity of such distributed systems is comparable to that of the largest supercomputers.
>
> However, these machines are connected via **internet-grade networks**, which are significantly slower (especially in terms of latency) and more failure-prone than server-grade infrastructure. As a result, existing distributed training methods that require **continuous high-throughput data exchange** are generally **incompatible** with this setup.
>
> ---
>
> ### 3. Long-context training is essential for competitiveness and scientific impact
>
> If crowdsourced models are to be **competitive with centralized LLMs**, they must be able to model **long-range dependencies**. Long-context capability is crucial for richer language modeling and better long-form reasoning, as well as other scientific domains.
>
> ---
>
> ### 4. Our work fills a critical gap
>
> Previous research has explored communication-efficient **data-parallel** [4, 5] and **model-parallel** [1,6, 7] strategies. However, **no prior work has enabled long-context training in bandwidth-constrained, decentralized environments**.
>
> Our method is the first to support **communication-efficient context parallelism**, making it feasible to train large, long-context models across **volunteer devices connected via the Internet**. We believe this opens the door to more inclusive LLM research and broader scientific applications beyond centralized infrastructure.
>
> ---
>
> ## Context parallel introduces frequent communication operations. Pipeline parallel might seem to be a better choice for low-bandwidth training with a much smaller number of communication volumes and frequencies.
>
> We would like to clarify that pipeline parallel allows splitting transformer blocks into devices, thereby enabling training large parameter scale models over a set of GPUs. However, it would not allow models to be trained with long context. For instance, the O(BN^2) memory complexity of the attention alone would consume 4154GB (include calculation) for a single transformer layer, for a model trained with pipeline parallel, which surpasses the most high-end single GPU capacity available even in most advanced data centers. This is why context parallel approaches have been proposed even in centralized settings. This need is more dire in decentralized volunteer settings since the lack of avaiability of high end GPUs.
>
> ## Context parallel introduces frequent communication operations. Pipeline parallel might seem to be a better choice for low-bandwidth training with a much smaller number of communication volumes and frequencies.
>
> We appreciate the reviewer’s point and would like to clarify the distinction between **pipeline parallelism** and **context parallelism** in this setting.
>
> Pipeline parallelism enables partitioning transformer blocks across devices, which is useful for **scaling model size** across limited GPU memory. However, **it does not address the challenge of long-context training**. Specifically, attention mechanisms incur **quadratic memory growth** with respect to the context length, $\mathcal{O}(B N^2)$, where $B$ is the batch size and $N$ is the sequence length.
>
> As a concrete example, consider a single attention layer in a model with:
> - Batch size $B = 32$
> - Sequence length $N = 132 \text{K}$
>
> The memory requirement just for the attention score matrix would be:
> $$
> B \times N^2 \times \text{dtype size} = 8 \times (132{,}000)^2 \times 4 \text{ bytes} \approx \mathbf{4.15\text{TB}}
> $$
>
> This **far exceeds the memory of even the largest available GPUs**, and such memory pressure applies to *every stage* in pipeline parallelism, as each stage needs to hold the full context during its local computation. Hence, pipeline parallelism alone becomes infeasible for training long-context models.
>
> This is exactly why **context-parallel approaches** have been proposed and adopted, even in centralized settings [8, 9]. The challenge is even **more pressing in decentralized environments**, where high-end GPUs are scarce and memory constraints are tighter. Our work makes context parallelism feasible under **internet-grade bandwidth**, allowing long-context models to be trained over distributed, volunteer devices.
>
> ## The surprising effectiveness of extremely low-dimensional rotations is acknowledged but not fully understood.
>
> Although a complete theoretical account is still an open problem, our observations align with a growing body of work indicating that large‑scale models possess wide, flat, and highly connected low‑loss regions in their optimization landscapes. Over‑parameterized networks often admit many near‑optimal local minima that are linearly connected [12, 13]. In this regime, the landscape resembles a web of flat valleys containing multiple solutions of nearly identical training, and frequently test, performance, so first‑order optimizers naturally converge within this region.
>
> Within such flat valleys, even low‑rank perturbations, for example, a rank‑1 rotation, can move the model to another point of equal loss. Recent analyses of permutation‑aligned weight spaces [10] show that independently fine‑tuned billion‑parameter models remain linearly mode‑connected after weight matching. Practical tools like SLERP, MergeKit [11], and model soups exploit this connectivity, merging models via low‑rank updates or simple averaging with minimal loss increase.
>
> Viewed through this lens, the effectiveness of setting  k=1 reflects traversal along a dominant direction inside the connected low‑loss region. Because many pre‑trained representations are already highly linearly separable, a single rotation can be sufficient to realign the features most critical for the downstream task.
>
> We will add a note of the above in supplementary. Thank you.
>
> ## The paper lacks comparison with other attention sparsification methods.
>
> We already compare with BigBird (see Fig. 5) which is performing sparse attention. If the reviewer can provide more citations, we will try our best to provide comparisons during the discussion period.
>
> [1] - SWARM Parallelism: Training Large Models Can Be Surprisingly Communication-Efficient
>
> [2] - Folding@home and genome@home: Using distributed computing to tackle previously intractable problems in computational biology
>
> [3] - Atlas@ home: harnessing volunteer computing for hep
>
> [4] - DiLoCo: Distributed Low-Communication Training of Language Models
>
> [5] - Photon: Federated LLM Pre-Training
>
> [6] - Protocol Models: Scaling Decentralized Training with Communication-Efficient Model Parallelism
>
> [7] - Beyond Top-K: Structured Sparsification for Compression in Pipeline Parallel
>
> [8] - Ring Attention with Blockwise Transformers for Near-Infinite Context
>
> [9] - DeepSpeed-Ulysses
>
> [10] -  Analysis of Linear Mode Connectivity via Permutation-Based Weight Matching
>
> [11] - Activation-Informed Merging of Large Language Models
>
> [12] - Loss Surfaces, Mode Connectivity, and Fast Ensembling of DNNs
>
> [13] - Essentially No Barriers in Neural Network Energy Landscape

---

### Official Review · Reviewer_2JUi · 2025-07-03

**Clarity:** 2
**Significance:** 2
**Originality:** 3
**Rating:** 4
**Confidence:** 4

**Summary:**

The paper addresses a critical and previously unexplored bottleneck in training large language models (LLMs): the immense communication overhead of Context Parallelism (CP) in decentralized, low-bandwidth environments. Standard CP requires each device to broadcast its local key (K) and value (V) activation matrices to all other devices, which is prohibitively expensive over commodity internet connections. The authors' core contribution is a novel and highly effective compression method for these activations. The method is motivated by the empirical observation that the K and V activation matrices in large transformers have an intrinsically low-rank structure. The proposed solution leverages this by: Projecting the high-dimensional K and V activations onto a low-dimensional subspace, drastically reducing the amount of data to be communicated. Dynamically adapting this subspace for each input chunk via a lightweight, learnable rotation. This provides flexibility without sacrificing compression. Using an efficient reparameterization of these rotations based on Lie groups, which avoids costly matrix operations (like SVD/QR) and makes the method computationally cheap. Introducing a mechanism to "unplug" the auxiliary compression components after training, resulting in a standard transformer architecture at inference time with no overhead.

**Questions:**

1. Could you provide more details on the warm-up stage? How was the duration (<500 iterations) and reduced context length determined? How sensitive is the final performance to these choices?
2. The success with a single rotation parameter (k=1) is fascinating. Do you have a hypothesis for why this highly constrained adaptation works so well? Have you identified any failure cases or do you foresee scenarios (e.g., highly multi-modal data) where a larger k would be necessary?
3. Figure 4 shows that late unplugging is effective. Is there a clear, automatable heuristic for determining the optimal point in training to remove the projection components? Does the subsequent "polishing" fine-tuning pass require a specific learning rate schedule to avoid disrupting the learned weights?
4. The paper uses 98% and 95% compression for K and V respectively. Have you explored the trade-off curve between the compression rate (i.e., the rank r) and the final model perplexity? It would be interesting to see the point at which performance begins to degrade meaningfully.
5. The proposed method compresses activations. Could it be productively combined with other techniques, such as weight quantization or gradient compression (for the infrequent weight syncs), to achieve even greater efficiency in a fully decentralized setting?

**Ethical Concerns:**

["NO or VERY MINOR ethics concerns only"]

**Final Justification:**

questions answered.

**Limitations:**

see questions.

**Quality:**

2

**Strengths And Weaknesses:**

Strengths
1. The paper tackles a highly relevant and timely problem. As context windows in LLMs expand beyond 100K tokens, CP is becoming essential. However, its application has been confined to expensive, centralized data centers. This paper is the first to propose a viable solution for CP in decentralized settings, potentially democratizing the training of long-context LLMs.
2. The method is built upon a simple yet powerful insight: the low-rank nature of attention activations. This is convincingly demonstrated in Figure 1 with an analysis of LLaMA-70B. This strong empirical grounding makes the subsequent methodological choices feel well-justified and intuitive.
3. The technical approach is very clever. The authors build a logical progression from a simple low-rank factorization to a more robust solution. The reparameterization of the projection subspace U via a fixed basis Ū and a learnable rotation R(θ) is an elegant way to maintain expressive power while making optimization tractable (avoiding expensive Riemannian gradient steps). The use of a low-dimensional Lie subgroup to parameterize R(θ) is a principled way to control the communication-flexibility trade-off.
4. The results presented are outstanding and strongly support the paper's claims. Performance Parity (Figure 2): Matching the wall-clock performance of a 100Gbps system on a 300Mbps link is a remarkable achievement and the most compelling result in the paper. Scalability (Figure 3): The method is shown to scale effectively to a 3B parameter model and integrates seamlessly with other parallelism techniques like Pipeline Parallelism. Ablations and Baselines (Table 2, Figure 5): The authors conduct thorough ablation studies that validate their design choices (e.g., the importance of the warm-start, the adequacy of the 2nd-order approximation). The comparisons against sensible baselines like quantization and sparsification clearly demonstrate the superiority of the proposed approach.

Weaknesses
1. The method relies on a "short, uncompressed warm-up phase" to establish the initial subspace basis Ū. This step is crucial, as the ablations show that performance degrades significantly without it ("No warm start" in Table 2). However, the details are sparse. The sensitivity to the duration and context length of this phase is not explored, making it seem somewhat heuristic. A more detailed analysis or guideline on setting up this phase would strengthen the paper.
2. The paper finds that a rotation parameterized by a single scalar (k=1) is sufficient. This is a surprising and powerful result for communication efficiency, but the paper offers little intuition as to why such a constrained, planar rotation is sufficient to adapt the subspace effectively across diverse data chunks. Is this a fundamental property, or could it be a limitation on more complex tasks or architectures?
3. The experiments are large-scale and computationally expensive, so this is understandable. However, the results are based on single training runs. As acknowledged in the paper's checklist, there are no error bars or confidence intervals to assess the variance of the results (e.g., due to different random seeds for initialization).

---

> ### Author Rebuttal · Authors · 2025-07-30
>
> We appreciate the reviewer’s constructive feedback and their encouraging remarks that our work tackles “a highly relevant and timely problem,” is “strongly grounded empirically,” employs a “clever technical approach,” and delivers “outstanding results.” Below, we address each of the reviewer’s questions in turn.
>
> ## 1) Could you provide more details on the warm-up stage? How sensitive is the final performance to these choices?
>
> Thank you for the thoughtful question. We clarify that our warm-up strategy is **hyperparameter-agnostic** and **aligns with standard context-length warm-up practices** used in modern LLM training.
>
>   - **Fixed warm-up configuration across all settings:** In all our experiments, we applied a **fixed warm-up phase of 500 steps** with a 2K context length, regardless of model size (ranging from 800M to 3B parameters), number of layers (8 to 32), or final context length (132K in the main paper; 256K in Fig. 13 of the supplementary). This includes all major results reported (e.g., Fig. 10 for 3B, Fig. 12 for 2.56B). Importantly, this warm-up setup is also used in matched uncompressed baselines to ensure a fair comparison. Across these varied settings, we did not observe significant sensitivity to the warm-up duration. The fixed schedule worked consistently well and demonstrates the stability of warm-up phase across scales.
>
>   - **Observed insensitivity to warm-up duration:**- We observed that the model maintains comparable performance even with a shorter warm-up of 300 steps. In practice, we adopt 500 steps as a conservative default.  In addition, we conducted an ablation study varying the warm-up duration to further address the reviewer’s concern. We evaluated the PPL on FW, with results presented below. This ablation highlights the robust nature of our warm-up, particularly when compared to the more complex scheduling strategies (learning rate, batch size, context length) often used in large-scale LLM pre-training. **Please note that perplexity is calculated as the exponential of the validation loss, so variations at the decimal level correspond to very small differences** (mostly the third decimal point in the validation loss).This indicates that performance remains stable after 300 warm-up steps.
>
> | Warm-up Steps | Perplexity |
> |-|-|
> | - | 26.63      |
> | 100 | 24.44      |
> | 300| 22.66      |
> | 500 | 22.64      |
> | 1000 | 22.87      |
> | 2000 | 22.64      |
> | 5000 | 22.71      |
>
>   - **Context-length warm-up is a well-established practice in frontier model training:** Note that context-length warm-up has become a standard component of many large-scale LLM training recipes, and our approach can be seamlessly fit into them.
>     - **Llama 3 400B** progressively increases context length in six stages, from 8K to 128K tokens [1].
>     - **Qwen 2** gradually grows the sequence length from 4K to 256K tokens [2].
>     - **Hugging Face SmolLM 3** follows a multi-stage progression from 4K to 32K to 64K tokens [3].
>     - **Li et al.** show that sequence-length warm-up helps reduce gradient variance, enabling GPT-3 to train with 8× larger batch sizes and 40× higher learning rates [4].
>
> Our own strategy, training briefly at a shorter context length and computing $\bar{U}$ before transitioning to the full sequence length, aligns well with these practices. Notably, our warm-up phase is much lighter-weight: we use only 500 steps at the reduced length, significantly fewer than the thousands of iterations employed in the examples above. This highlights the practicality and efficiency of our approach.
>
> Moreover, we would like to point out that modern LLM pre-training already incorporates multiple warm-up mechanisms, including those for learning rate, batch size, and context length, each of which has much stronger effects on the final convergence. In comparison, our method is both simpler and less sensitive to hyperparameter tuning.
>
> **If the reviewer would like to see additional experiments to further clarify this point, we would kindly invite them to indicate so in their response. We will make every effort to provide the requested results within the discussion period.**
>
> ## 2) The success with (k=1) is fascinating. Do you have a hypothesis for why?
>
> We agree with the reviewer that the effectiveness of a single rotation parameter (k = 1) is fascinating. While a full theoretical explanation remains an open question, we believe this behavior aligns with a growing body of evidence that the loss landscapes of large models contain wide, flat, and highly connected regions of low loss, resulting in equally good multiple local minima.
>
> In particular, it is well-established that overparameterized networks often admit many near-optimal local minima that are linearly connected [7,8]. In this regime, optimisation landscape becomes a high‑dimensional web of connected, flat valleys containing multiple solutions with near‑identical performance, and first‑order methods fall into this region. This implies that even low-rank updates (such as a rank-1 rotation) can be sufficient to move within a flat valley of good solutions without increasing loss. Relatedly, recent analyses of permutation-aligned weight spaces [5] show that large models fine-tuned independently remain linearly mode-connected after weight matching. Practical techniques such as SLERP, MergeKit [6], and model soups further exploit this connectivity, achieving successful model merging via low-rank updates or averaging.
>
> From this view, the success of k = 1 can be interpreted as navigating a dominant direction in this connected low-loss region. Since many pre-trained representations are already highly linearly separable, a single rotation may suffice to realign critical features.
>
> ## 3) Have you identified any failure cases or do you foresee scenarios where a larger k would be necessary?
>
> We appreciate the reviewer's concern. We did not observe any failure cases in our experiments. However, if a larger k becomes necessary, a router could select among several projection heads with different *k* based on activation statistics, trading bandwidth for expressiveness, an avenue for future work. Note that because our method addresses only the comm bottleneck, it can be combined with expressiveness-oriented methods (MoE routers, dynamic sparsity, low-rank adapters). Should *k = 1* prove limiting, such hybrids can restore capacity while retaining communication savings.
>
> ## 4) Is there a automatable heuristic for determining the optimal point in training to remove the projection components?
>
> Thank you for the question. In our experiments, we found that removing the projection components around the last 30%-5% range of training consistently results in stable performance across diverse settings. Based on this observation, we recommend using the last 30%-5% range of training as a simple and effective heuristic, and we will make this recommendation explicit in the revised version of the paper.
>
> Further, as in Fig. 4, even when the projection components are removed earlier in training, the model is able to recover quickly, indicating a degree of robustness to the exact transition point. Nonetheless, the last 30% heuristic offers a practical, safer, and an automatable guideline. We present the below ablation on FW to further illustrate this (total steps 10k). Variations correspond to the third decimal point in the validation loss in most cases.
>
> | Unplugging Step | PPL|
> |-|-|
> | Not unplugged | 22.64 |
> | 5k| 22.69|
> | 7k| 22.64|
> | 8k| 22.71|
> | 9k| 22.71|
> | 9.5k| 22.77|
> |9.9k | 23.12|
>
> ## 5) Have you explored the trade-off curve between the compression rate  and the final PPL?
>
> Thank you for the thoughtful question. We have already provided an ablation in Table 2 showing the drop in performance when V-compression is increased to 98% and K-compression to 99%. Further, we present a more granular ablation on the FW to show the trade-off between compression rate and final model perplexity.
>
> The baseline perplexity w/o compression is 22.65.
>
> | K| V| PPL|
> |-|-|-|
> | 98%           | 95%            | 22.64       |
> | 98%           | 96%            | 22.69       |
> | 98%           | 97%            | 22.69       |
> | 98%           | 98%            | 22.74       |
> | 99%           | 95%            | 24.68       |
> | 99%           | 96%            | 24.71       |
> | 99%           | 97%            | 24.77       |
> | 99%           | 98%            | 25.23       |
> | 99%           | 99%            | 26.44       |
>
> We will include above results in the revised version.
>
> ## 6) Could the method be combined with weight quantization or gradient compression?
>
> We absolutely believe this is indeed possible! Our method is orthogonal to and fully compatible with other DDP-style compression techniques. Since we targets activation and activation-gradient compression, it can be directly combined with weight quantization methods that reduce model size and memory footprint. These methods operate on different components of the training pipeline and can complement each other effectively. Further, our technique applies to context-parallel setups where activations are transmitted across layers. In contrast, gradient compression techniques like Top‑K, PowerSGD, or quantized all-reduce are designed for DDP settings, where gradients are exchanged between full-model replicas. We believe  that our method can be layered on top of DDP-style gradient compression, yielding compounded communication savings.
>
> [1]-The Llama 3 Herd of Models
>
> [2]-Qwen2.5-1M: Deploy Your Own Qwen with Context Length up to 1M Tokens
>
> [3]-SmolLM3: smol, multilingual, long-context reasoner
>
> [4]-The Stability-Efficiency Dilemma: Investigating Sequence Length Warmup for Training GPT Models
>
> [5]- Analysis of Linear Mode Connectivity via Permutation-Based Weight Matching
>
> [6]- Activation-Informed Merging of Large Language Models
>
> [7]-Loss Surfaces, Mode Connectivity, and Fast Ensembling of DNNs
>
> [8]-Essentially No Barriers in Neural Network Energy Landscape

---

> > ### Comment · Reviewer_2JUi · 2025-08-06
> >
> > Thanks for the answers, which have solved most of my questions. I would like to raise to a 4.

---

### Official Review · Reviewer_w6eq · 2025-07-03

**Clarity:** 2
**Significance:** 2
**Originality:** 2
**Rating:** 3
**Confidence:** 5

**Summary:**

The paper attempts to solve the efficiency problem of long-context pretraining under low-bandwidth connections situation. Concretely it studies compression to reduce the communication problem, incurred by ring-style sequence parallelism. It achieves scaling billion-parameter models with more than 100k on slow networks.

**Questions:**

(1) Is it possible that the author provides experiment result on larger models?  The convergence can be different for larger model (the paper pretrains a 800M model, where the reviewer suggest the scale should be > 7B to draw significant applicable conclusion), especially when the paper method is based on compression.
(2) Is it possible to show effectiveness on larger cluster? The current experiments are done on 8xA100, where typically models are pretrained at the scale of > 100 H100 GPUs.
(3) How does the method applies to Ulysses style sequence parallelism, which typically induces less communication?
(4) (Minor) The figure 3 and figure 4 ratio are different. Could the author align them for a more compact view of the paper?

**Ethical Concerns:**

["NO or VERY MINOR ethics concerns only"]

**Final Justification:**

The paper is generally good, but as the original review points out, larger scale experiments are needed to validate the method.

**Limitations:**

Yes.

**Paper Formatting Concerns:**

No major issues.

**Quality:**

2

**Strengths And Weaknesses:**

Strengths:
-  The problem on ring-style sequence parallelism is significant.
- The paper is well written and easy to follow.

Weakness:
- The convergence is done on a small model, compared to the models mentioned in the introduction, e.g. Llama4.  The convergence can behave differently.
- It does not studies Ulysses style sequence parallelism, where the communication is less a problem, compared to ring-style.

---

> ### Author Rebuttal · Authors · 2025-07-30
>
> We thank the reviewer for identifying the significance of our work and for the constructive feedback. Please find our answers below.
>
> ## 1)  Is it possible to show effectiveness on larger cluster with a bigger model than 800M? The current experiments are done on 8xA100, where typically models are pretrained at the scale of > 100 H100 GPUs.
>
> We appreciate the reviewer’s concern. As noted in the paper (Figures 3, 10 (supp) and 12(supp)), we have already evaluated our method on **2.5B** and **3B** parameter models using 32 A100 GPUs. In both cases, our method maintains convergence while achieving substantial communication savings, **yielding approximately 20× improvement in tokens per second (TPS)** compared to the uncompressed baselines.
>
> While we currently lack access to large H100 GPU clusters to train larger models at massive scale, we see no fundamental limitation in the method that would prevent it from scaling. Our experiments already operate at the billion-parameter level, and the design of the algorithm, particularly its low overhead and compatibility with standard training procedures, suggests it would extend naturally to larger models and clusters.
>
> We hope our results offer a strong foundation for future work on communication-efficient context parallelism at larger scales, and we welcome further exploration of the method in such settings.
>
> ## 2) How does the method applies to Ulysses style sequence parallelism?
>
> We thank the reviewer for pointing out Ulysses. We now clarify why Ulysses is not applicable in decentralized settings.
>
> ### Why Ulysses is **not** applicable in decentralized settings?
>
> **Ulysses communication savings *only* stems from modern GPU clusters with intra-node NVSwitch interconnect and inter-node fat tree IB topology**. Specifically, Ulysses's communication savings are entirely dependent  on intra-node NVSwitch crossbars and an inter-node 1 : 1 fat-tree InfiniBand fabric that ensures the communication volume transmitted per link for an all-to-all for aggregate message of size M over P GPUs is M/P.  In decentralised setting, such topologies do not exist and peers communicate over heterogeneous, asymmetric residential links that traverse multiple ISP hops and NATs. Further, in Ulysses, every Transformer block performs a synchronous NCCL _all-to-all_ of QKV and context vectors. In decentralized settings, this is also not possible.
>
> We would also like to highlight that our method is **not restricted to ring-style collectives and is fully compatible with all-to-all gather and other communication patterns**. Since it only compresses the exchanged K and V vectors, the approach is agnostic to the underlying communication protocol and can be seamlessly integrated with any collective communication strategy.
>
> ### Can our method improve *centralized cluster* communication efficiency?
>
> As we explained above, **Ulysses cannot be applied in a decentralized environment** due to its structural reliance on synchrony and fabric-level assumptions. Because these assumptions are architectural, Ulysses cannot be ported by pure algorithmic tweaks.
>
> Interestingly, we emphasize that the **reverse** is possible. That is, if deployed in a centralized cluster setting, it is possible that our method can be **overlaid on Ulysses** to reduce traffic significantly, making it attractive for centralized, large-scale, long-context models. Below, we compare the theoretical communication cost per Transformer block in this case. For a context length `N` and an embedding dimension `h`,
>
> | Scheme | Data moved per GPU (forward **or** backward) | If we **overlay our 95 % QKV compression** |
> |--------|--------------------------------------------------|-------------------------------------------|
> | **Ulysses**| `3 · N · h` bytes  (Q, K, V) + `1 · N · h` bytes  (context)  → **`4 N h`** | **`0.05 × 3 N h + 1 N h ≈ 1.15 N h`** |
> | **Our Context-Parallel (CP)** | **`≈ 0.1 N h`** bytes  | unchanged |
>
>
> **Summary** Even in a *central* cluster our compression reduces Ulysses traffic by **≈ 3.5 ×**, i.e.,  (`4 N h → 1.15 N h`).  In contrast, Our CP is *already* an order-of-magnitude lower (`0.1 N h`). Below, we demonstrate a toy scenario with some practical numbers. For example considering a config, `N = 4 096`, `h = 4 096` we get,
>
> | Variant | Bytes / GPU / block |
> |---------|---------------------|
> |Ulysses| 100 MB |
> | Ulysses + ours | 26 MB |
> | **Ours** | 2.5 MB |
>
> Therefore, although Ulysses is an elegant solution for data-centre long-context training, but its dependence on  intra-node NVSwitch interconnects and inter-node fat tree IB topology-based fabrics makes it unusable for the decentralised, bandwidth-limited setting our work targets.  On the contrary, **our  compression shrinks Ulysses traffic significantly on *central* clusters and could indeed be combined with Ulysses there**.  Consequently, we do not view Ulysses as a realistic option for decentralised training, whereas our context-parallel method is expressly designed for, and validated in, such resource constrained setting.
>
> ## 3) (Minor) The figure 3 and figure 4 ratio are different. Could the author align them for a more compact view of the paper?
>
> We thank the reviewer for pointing this out and we will correct this in the revised version.

---

> ### Comment · Reviewer_w6eq · 2025-08-01
>
> Thank you for the rebuttal! Really appreciate it!
>
> (1) I see the result from 2.5B and 3B now.
> (2) Ulysses has limitation in de-centeralized setting.
>
> However, as my original review points out, since the paper targets for long context-pretraining, it would be better for the paper to directly test on more current generation of pre-trained models, e.g. >7B, since the convergence could be different, and with the final accuracy number.
>
> I understand that the resources are hard to get, but maybe the author could find alternative reference to convince the audience on the accuracy: e.g. some mid-training scnerios that focuses on long-context extension (which doesn't require the author to train from scratch), and validate the accuracy against those papers.
>
> And unfortunately, I have to keep my scores due to this major concern.

---

> ### Author Response · Authors · 2025-08-02
>
> We sincerely thank the reviewer for the continued engagement and thoughtful suggestions.
>
> We fully respect the reviewer’s position regarding evaluation on >7B models. We agree that larger-scale evaluations would further strengthen the empirical case.
>
> That said, we would like to respectfully note that access to compute at that scale remains a significant barrier for many researchers. Training a 7B model, even under a very conservative batch sizes, would typically require at least 50 H100 GPUs, which places it well beyond the reach of most academic research groups. We hope the community continues to recognize the value of contributions that push *algorithmic scalability*, even when full-scale validation is not immediately feasible.
>
> We also want to emphasize the **practical relevance of 1–3B scale models**. Many widely-used open-source models, such as **OLMo-1B, Phi-2, Qwen-1.8B, SmolLM3B, and Falcon-RW-1B**, fall in this range and are actively deployed in both research and production due to their favorable latency and memory trade-offs. As such, our method offers utility even without extrapolation to larger scales.
>
> Moreover, the **literature provides strong evidence that larger models are often easier to compress** due to higher redundancy and smoother loss landscapes:
> - **Brownet al.** shows that the compressibility improves with model scale.
> - **Li et al.** larger Transformers are more robust to compression  than smaller ones
> - **Zhu et al.** found that a large models can be pruned down more effectively compared to small models
>
> These works demonstrate that compression becomes **more effective** at scale, supporting the claim that our method, which performs well at 1–3B scale, is likely to retain or even improve its effectiveness on 7B+ models.
>
> We appreciate the reviewer’s suggestion to explore long-context extension settings mid-training, and will consider this in future work. We are grateful for your thoughtful engagement.
>
> Brown et al.: How does LLM compression affect weight exfilteration attacks
>
> Li et al.: Train Large, Then Compress: Rethinking Model Size for Efficient Training and Inference of Transformers
>
> Zhu et al.: To prune, or not to prune: exploring the efficacy of pruning for model compression

---

### Official Review · Reviewer_sGpa · 2025-07-08

**Clarity:** 3
**Significance:** 3
**Originality:** 2
**Rating:** 5
**Confidence:** 5

**Summary:**

This paper introduces a compression method enabling efficient context-parallel training of large language models across decentralized devices with limited bandwidth. By dynamically exploiting the low-rank structure of attention activations through learned mixtures of subspaces, the method achieves over 95% compression without sacrificing model convergence or performance. Experiments demonstrate successful training at scale, matching centralized high-bandwidth systems in convergence speed.

**Questions:**

How does the method proposed relate to PowerGossip (https://arxiv.org/abs/2008.01425) and Photon (https://arxiv.org/abs/2411.02908)?

You mention an initial warm-up phase is crucial to your method; how sensitive is your approach to variations in this phase (such as reduced length or interrupted warm-up), and have you explored strategies to mitigate dependency on it in dynamic decentralized environments?

Could you clarify how robust your method is when the low-rank assumption of attention activations is violated or weaker than expected? What happens to performance and convergence in such scenarios?

**Ethical Concerns:**

["NO or VERY MINOR ethics concerns only"]

**Final Justification:**

Below is my original justification part way through the process. After the authors clarified how the additional material discussed during the rebuttal period would appear in the paper, I have become more comfortable with the work. And so have a raised my score.
----

I have considered the reply from the authors. I think the information it provides provides important additional context and observations for the main paper. At this stage I haven't received word if they would include the rebuttal material into the main paper, and so have asked about it. If they can more clearly describe how the final paper will look, then I might consider increasing my score further. I am already fairly positive. I plan to edit this, if I receive a reply from them.

**Limitations:**

Nothing too strong to push on.

**Paper Formatting Concerns:**

I'm ok with the formatting.

**Quality:**

2

**Strengths And Weaknesses:**

I really liked how effectively they tackle the major bottleneck in decentralized training, namely communication. Reducing bandwidth usage by over 95% while preserving accuracy is seriously impressive. The method is also flexible, integrating seamlessly into existing transformer architectures, so you can drop their special components after training and just deploy a normal model. Plus, they show practical scaling to billion-parameter models, which is exactly what decentralized setups need.

On the downside, though, the technique relies heavily on an initial "warm-up" phase, which might be tricky in unpredictable real-world setups. Another concern is that the theoretical guarantees lean on assumptions (like low-rank activations) that might not hold perfectly across all tasks or domains. Finally, while compression rates are great, their rotation-based subspace method introduces extra computational complexity, even though they've tried to minimize it.

---

> ### Author Rebuttal · Authors · 2025-07-30
>
> We sincerely thank the reviewer for the positive and encouraging feedback, with remarks such as our work being “seriously impressive,” the method being “flexible,” “integrating seamlessly into existing transformer architectures,” and demonstrating “practical scaling to billion-parameter models.” Below, we address the specific questions raised.
>
> ## 1) Initial "warm-up" phase, which might be tricky
>
> We thank the reviewer for the thoughtful question. Below, we note that our warm‑up phase is **agnostic across different hyperparameters** and **can be seamlessly integrated into existing context‑length warm‑up practices.**
>
> - **We use a fixed warm- setting across all settings:**  In our experiments, we used a **fixed warm-up phase of 500 steps with 2k context length across all setups**, regardless of model size (from 800M to 3B parameters), number of layers (8 to 32), or final context length (132K in the main paper and 256K in Fig. 13 of the supplementary). This includes models shown in Fig. 10 (3B), Fig. 12 (2.56B), and others. We also ensured that the warm-up configuration was consistent with matched uncompressed baselines. Based on these results, **we do not observe strong sensitivity of our method to the length of the warm-up phase across varying model scales or configurations.**
>
> - **Context length warm-up is already a standard practice across many frontier models:**
>
>   - We would also like to emphasize that context‑length warm‑up is now standard in many frontier‑model training recipes:
>     - **Llama 3 400B** gradually expands the context window in **six stages**, from **8 K → 128 K** tokens [1].
>     - **Qwen 2** increases sequence length from **4 K → 256 K** tokens [2].
>     - **Hugging Face SmolLM 3** steps from **4 K → 32 K → 64 K** tokens [3].
>     - **Li et al.** demonstrate that sequence‑length warm‑up mitigates gradient variance, allowing GPT‑3 to train with **8 × larger batch sizes** and **40 × higher learning rates** [4].
>
> Our own schedule (training briefly at a shorter context length and computing $\bar{U}$ before switching to the final length) fits naturally within these prevailing practices. Moreover, our warm‑up is **extremely lightweight** and **less imposing** than above schedules: we use only 500 steps at the shorter length, far fewer than the thousands of iterations reported in the works above, highlighting the practicality of our recipe.
>
> We would also like to emphasize that modern LLM pre‑training already employs multiple, often intricate warm‑up schedules, covering learning rate, batch size, and context length, whereas our recipe is comparatively simple and far less sensitive to hyperparameter choices.
>
> ## 2) How sensitive is your approach to variations in the warm up phase
>
> In addition to the above-mentioned points, to further address the reviewer’s concern, we conducted an ablation study varying the warm-up duration and evaluated the resulting perplexity on the FineWeb dataset. The results are shown below. As demonstrated, even with a reduced warm-up of 300 steps, the model achieves comparable performance, indicating no significant degradation. In practice, we default to 500 steps to provide a safe and stable baseline. This study further emphasizes the lightweight and robust nature of our warm-up strategy, especially in contrast to the more elaborate scheduling mechanisms commonly employed in modern LLM pre-training.
>
> Please note that perplexity is computed as the exponential of validation loss, so differences in decimal points are minor and stable, indicating performance is stable after 300 warm-up steps.
>
> | Warm-up Steps | Perplexity |
> |---------------|------------|
> | -             | 26.63      |
> | 100           | 24.44      |
> | 300           | 22.66      |
> | 500           | 22.64      |
> | 1000          | 22.87      |
> | 2000          | 22.64      |
> | 5000          | 22.71      |
>
> We will include these results in the supplementary. **If the reviewer needs more experiments or clarifications, we would like to kindly ask the reviewer to mention them so we can try our best to provide them during the discussion period.** Thank you for the suggestion.
>
>
> ## 3) Relationship to Power Gossip and Photon
>
> Thank you for pointing us to these interesting works. We note that both papers operate in a data-parallel (DDP) setting, where each node holds a full model replica and exchanges weights or weight gradients to simulate a larger batch size. Note that this is a comparatively well studied domain, and is orthogonal to the unexplored decentralized context parallel setting that we explore.
>
> Power Gossip, for example, replaces synchronous all-to-all communication with gossip-style information exchange among neighboring replicas arranged in a mesh. Its key insight is that, when each replica trains independently via local SGD, the pairwise weight differences evolve in a low‑rank sub‑space, enabling them to efficiently compress the *weight differences* during gossip. For instance, assume $W_i^t$ is a weight matrix of replica $i$ at time $t$. It's update at time $t+1$ is denoted as,
>
> $$W_i^{t+1}  = W_i^t + \\sum M_{ij} \\mathcal{C} (W_i^t  - W_j^t ) - \\eta G$$
>
> where $\eta$ is the learning rate, $G$ is the local gradient matrix, $\\mathcal{C}$ is the low-rank compression, and $M_{ij}$ denotes weighting between the nodes i and j in the connectivity graph. This shows that the weight $(W_i^t  - W_j^t )$ at any particular time $t$ can be approximated with a low-rank matrix. Now, consider the following interesting upperbound.
>
> $$\text{rank}(W_i^t  - W_j^t ) \leq \text{rank}(W_i^t  ) + \text{rank}(W_j^t )$$
>
> This means, that if both $W_i^t $ and $W_j^t$ are low rank, their differences are likewise low‑rank: precisely the property Power Gossip exploits! This insight parallels our finding that attention projection matrices collapse to a low‑rank sub‑space during training. Note that however, our sub‑space compression is complementary to Power Gossip‑style techniques: while they compress weight differences in a DDP setting, we compress activations and activation gradients in context‑parallel settings. Together, these methods could further reduce communication in hybrid training setups.
>
> Photon also targets end‑to‑end LLM training in a DDP setting. Its communication savings stem primarily from infrequent gradient exchanges rather than from any explicit compression scheme. Such skip‑sync approaches are infeasible in context‑parallel pipelines, where activations must be transferred between nodes at every forward and backward pass. Nevertheless, like PowerSGD, these DDP‑style techniques are orthogonal to our method and could be combined with it in hybrid setups.
>
> Our work fills a gap in decentralised distributed training: while communication‑efficient DDP and model‑parallel techniques have been explored, we are, to our knowledge, the **first to focus specifically on communication‑efficient context parallelism.**
>
> ## 4) Could you clarify how robust your method is when the low-rank assumption of attention activations is violated or weaker than expected?
>
> Thank you for raising this point. We address it in three parts.
>
> **Empirical evidence across diverse frontier models:** We analysed attention activations in several open‑source LLM families (Llama, Qwen, and OLMo) spanning 3 B to 235 B parameters and trained on multilingual corpora (see Figs. 6–8). In every case the singular‑value spectrum is sharply concentrated. This suggests the low‑rank property is not an artefact of a single architecture or task, but a consistent outcome of large‑scale pre‑training. Further, this is a strong empirical proof that our method can produce competitive models.
>
> **Dynamic expressiveness:** Hypothetically, if a future setting required richer attention representations, one possible solution (among many other options) is to add a lightweight router that selects among multiple projection heads with different k values, conditioned on the activation statistics. This would allow the model to adaptively adjust expressiveness, with a trade-off on communication efficiency.  We leave such interesting explorations to future work.
>
> **Orthogonality to expressiveness‑enhancing techniques:** Our approach targets the communication bottleneck; it does not alter the functional form of the attention mechanism. Therefore it can be combined with methods that boost expressiveness, e.g., MoE routers, dynamic sparsity, or low‑rank adapters, should the low‑rank assumption ever become limiting.
>
> In summary, extensive empirical evidence shows the low‑rank assumption holds across today’s frontier models, and even if future tasks weaken this property, our scheme offers straightforward levers (larger k or adaptive routing) to maintain accuracy while still providing substantial bandwidth savings.
>
> [1] - The Llama 3 Herd of Models
>
> [2] - Qwen2.5-1M: Deploy Your Own Qwen with Context Length up to 1M Tokens
>
> [3] - SmolLM3: smol, multilingual, long-context reasoner
>
> [4] - The Stability-Efficiency Dilemma: Investigating Sequence Length Warmup for Training GPT Models

---

> ### Comment · Reviewer_sGpa · 2025-08-08
> **Thanks**
>
> Thank you again for the thoughtful and detailed rebuttal. I found the new experimental results and clarifications quite valuable. In particular the warm-up sensitivity analysis, the expanded discussion of low-rank structure across model families, and the positioning of your method relative to PowerGossip and Photon.
>
> Would it be possible to this rebuttal material into the final paper? One natural place would be to expand the background or related work sections to include the comparisons these other works, as well as any others you have seen. As they help clearly situate your contribution in the broader distributed training landscape.
>
> Similarly, the warm-up ablation and discussion of its robustness would be useful to reference in the main text, not just the appendix I think. Even if the full results still need to be placed in the appendix itself. This helps reassure readers that the method remains practical and resilient in varied settings. Likewise, the empirical evidence supporting the low-rank assumption could be briefly summarized in the main paper, with a reference to the figures and analysis in the appendix. These details strengthen your theoretical foundations and anticipate likely reader concerns.
>
> Overall, these additions would make the paper more self-contained and accessible, and I believe they would significantly enhance its clarity and impact. Thanks again for the clear rebuttal and additional context. Let me know your thoughts on these suggestions.

---

> > ### Author Response · Authors · 2025-08-09
> >
> > Thank you for the thoughtful and constructive follow-up. We completely agree that incorporating the additional material from the rebuttal will make the paper more self-contained, accessible, and impactful.
> >
> > Since NeurIPS allows one additional page in the camera-ready version (10 pages instead of 9), we will use this extra space to integrate the following into the main text:
> >
> > - **Expanded positioning in the distributed training landscape** – In **Section 1 (Introduction)** and **Section 4 (Related Work)**, we will include the new discussions on **PowerGossip** and **Photon** from the rebuttal, along with other relevant works. This will clarify how our method differs from and complements existing model-parallel and data-parallel communication-efficient approaches.
> >
> > - **Warm-up sensitivity analysis** – In experiments, we will summarize the warm-up ablation results and robustness discussion from the rebuttal.
> >
> > - **Empirical evidence for the low-rank assumption** – In **Section 2**, we will briefly summarize the cross-model analysis (LLaMA, Qwen, OLMo, plus ViTs) and link to the detailed spectra plots in the appendix. This will help address the common question of whether our low-rank premise holds across architectures and modalities.
> >
> > We agree that these changes will significantly improve clarity without disrupting the overall flow, and they can be implemented comfortably within the extra page allowance. We appreciate your encouragement to strengthen the paper in these directions, and we will ensure the final version reflects this feedback.

---

> > > ### Comment · Reviewer_sGpa · 2025-08-09
> > >
> > > Thanks for clarifying how you'll go about revising the paper with these new results and discussions. I appreciate time and care you have taken during the rebuttal phase. I'm happy for this paper to be published, and will be increasing my score.

---

### Decision · Program_Chairs · 2025-09-17

**Decision:**

Accept (poster)

**Comment:**

The rebuttal was quite successful here, and the motivation and new experimental data convinced most reviewrs. Two reviewers raised their scores to solid accepts. While one reviewer complained about lack of >7B experiments, another reviewer thought this was not grounds for rejection and other reviewers did not raise this concern (if we were to disregard this complaint, the paper would have a score above the suggested borderline range). There was still some concerns from one reviewer about motivation. Overall though the paper was praised for its novel and impactful solution to an important problem.